# Disparities in Suicide Mortality Between Indigenous and Non-Indigenous Populations in Southern Brazil (2010–2019)

**DOI:** 10.3390/ijerph22091313

**Published:** 2025-08-22

**Authors:** Thiago Fuentes Mestre, Fernando Castilho Pelloso, Deise Helena Pelloso Borghesan, Ana Carolina Jacinto Alarcao, Pedro Beraldo Borba, Vlaudimir Dias Marques, Paulo Acácio Egger, Kátia Biagio Fontes, Fernanda Cristina Coelho Musse, José Anderson Labbado, Elizabeth Amâncio de Souza da Silva Valsecchi, Jorge Luiz Lozinski Musse, Amanda Carina Coelho de Morais, Raissa Bocchi Pedroso, Sandra Marisa Pelloso, Maria Dalva de Barros Carvalho

**Affiliations:** 1Health Sciences Center, State University of Maringá—UEM, Maringá 87020-900, Brazil; vlaudimirdm@gmail.com (V.D.M.); paegger@uem.br (P.A.E.); andersonlabbado@hotmail.com (J.A.L.); jorgemusse@yahoo.com.br (J.L.L.M.); raissap@gmail.com (R.B.P.); smpelloso@gmail.com (S.M.P.); mdbcarvalho@gmail.com (M.D.d.B.C.); 2Municipal Health Department, Curitiba 80060-130, Brazil; fercaspell@gmail.com; 3Union of Catholic Colleges of Cuiabá, Várzea Grande 78070-200, Brazil; deisepelloso@hotmail.com; 4Departament of Psychology, Adventist Faculty of Paraná, Ivatuba 87130-000, Brazil; anacjalarcao@gmail.com; 5Department of Medicine, University of Marília—UNIMAR, São Paulo 17525-902, Brazil; bborbapedro@gmail.com; 6Departament of Nursing, Paranaense University—UNIPAR, Umuarama 87502-210, Brazil; katia.bf@gmail.com; 7Departament of Medicina, State University of Maringá—UEM, Maringá 87020-900, Brazil; fcoelho_med@hotmail.com; 8Departament of Nursing, State University of Maringá—UEM, Maringá 87020-900, Brazil; eassvalsecchi@uem.br; 9Prefeitura Municipal de Maringá, Maringá 87013-230, Brazil; amandacoelho_med@hotmail.com

**Keywords:** suicide, population groups, Indigenous people, time-series studies

## Abstract

The objective of this study was to evaluate the temporal trend of suicide deaths and the disparities in the occurrence of suicide death between Indigenous and non-Indigenous populations. This ecological study analyzed deaths from intentionally self-inflicted injuries in the southern region of Brazil (states of Paraná, Santa Catarina, and Rio Grande do Sul) from 2010 to 2019. The variables analyzed included annual frequency of events, age, sex, marital status, and education level. Descriptive analysis, association measures, and verification of temporal trends were performed. The average age-standardized suicide mortality rate for both populations was approximately 9.0 per 100,000 inhabitants, with a predominance among males (80%), single individuals (65%), and youth (37%). When only the state of Paraná was considered, the mortality rate during the period was 12.41 among the Indigenous population versus 6.94 per 100,000 inhabitants in the non-Indigenous population. In this state, the probability of suicide death was found to be almost twice as high among Indigenous men and women, with 13 times greater odds of death among Indigenous children and youth. A statistically significant temporal increase in the number of cases was observed only in the non-Indigenous population. An annual seasonal pattern of events among Indigenous individuals was suggested. The findings indicate a high suicide rate among the Indigenous population, particularly in Paraná, especially among young, single individuals, with a growing trend over the years.

## 1. Introduction

The World Health Organization (WHO) estimates that more than 800,000 people die by suicide each year, equivalent to one person every 40 s. In 2019, there were approximately 703,000 suicide deaths—exceeding the number of deaths caused by homicide or war. This corresponds to a global age-standardized suicide rate of 9.0 per 100,000 inhabitants [1].

In Brazil, the age-standardized suicide rate for 2019 was estimated at 6.4 per 100,000 inhabitants, with significant differences between sexes: 2.8 among women and 10.3 among men [1]. These disparities are not limited to sex; other variables, particularly socioeconomic and cultural factors, are also important, especially among vulnerable groups such as Indigenous peoples.

Researchers have urged the WHO to address what they describe as a universal phenomenon of elevated suicide rates among Indigenous populations [2]. Notable examples include alleged epidemics in Inuit communities in North America, Amazonian tribes in the Guianas, and Aboriginal peoples in Australia [2].

In Brazil, several studies have documented increasing suicide rates among the Guarani Kaiowá and Guarani Nandeva Indigenous communities in the state of Mato Grosso do Sul (central-west region). From 2000 to 2003, 194 suicides were reported, representing 50% of all deaths due to external causes and about 10% of total deaths in the studied population [3]—which is well above expected levels for the general population.

A systematic review examined, in addition to sociodemographic elements, risk factors such as poverty, family and cultural disorganization, alcohol problems, and land conflicts [4]. The review included data from populations in the north and central-west regions (Terena, Kadiweu, Guató, Ofaié-Xavante, Guarani, Guarani-Kaiowá, and Guarani-Nandeva), but offered no information on Indigenous populations from the south of Brazil [4].

The southern region of Brazil comprises the states of Paraná, Santa Catarina, and Rio Grande do Sul, where approximately 75,000 individuals self-identify as Indigenous [5]. There are four main ethnic groups: Guarani, Kaingang, Xokleng, and Xetá [6]. The Guarani are the second largest Indigenous group in Brazil, followed by the Kaingang [7]. The last two reside in territories belonging to the others and are on the verge of cultural extinction [6].

Given the importance of the topic, the scarcity of studies, and the vulnerability and invisibility of these populations, this study aimed to analyze the temporal trend in deaths from self-inflicted injuries among Indigenous and non-Indigenous populations in southern Brazil, assessing disparities across different sociocultural variables.

## 2. Materials and Methods

This is an ecological time-series study analyzing trends in suicide mortality in the southern region of Brazil.

Suicide data were obtained from the Mortality Information System (SIM), maintained by the Health Surveillance Secretariat (SVS), using the TABNET tool (web version) of the Department of Informatics of the Brazilian Unified Health System (DATASUS). Within this tool, in the “Vital Statistics” section, data on deaths from external causes were collected for the region of interest from 2010 to 2019. The group “Intentionally self-inflicted injuries” in the International Statistical Classification of Diseases and Related Health Problems (ICD-10) corresponds to codes X60 through X84. The analyzed variables include the number of events, sex, age, marital status, and educational attainment. The comparison was made between the Indigenous population and the non-Indigenous population of each state. An Indigenous person is anyone who identifies as and is recognized as belonging to Indigenous people—based on self-identification, ancestry, or cultural and territorial ties to an original population. National surveys such as the Census use the “race/color” question to classify individuals, and those who self-declare as Indigenous are categorized accordingly.

The national population census conducted by the Brazilian Institute of Geography and Statistics (IBGE) was disrupted due to the COVID-19 pandemic in 2020. The Indigenous population was estimated using the geometric growth rate of the general population. The Indigenous population for each year between 2010 and 2019 was estimated by applying the proportion of Indigenous individuals identified in the 2010 national census to the annual general population estimates provided by the Brazilian Institute of Geography and Statistics (IBGE). This method assumes a constant Indigenous share over time and allows for the calculation of yearly Indigenous population figures based on projected total population growth. Age- and sex-specific Indigenous population estimates were based on their relative share in the total population, following methodologies used in official studies [8]. Mortality data were then age-standardized using the direct standardization method, based on the WHO standard population [9]. Standardization consisted of a weighted average of age-specific mortality rates for the population of interest, reflecting the expected values if the age distributions were identical.

Descriptive analyses were conducted to generate frequency tables and graphs characterizing the study population. Absolute and relative frequencies were used for categorical variables.

To assess the presence of trends in suicide rate series, the Mann–Kendall trend test was used, with a significance level of α = 0.05. Trend verification also included graphical analysis of each time series and linear trend lines generated through linear regression models. This approach was chosen due to the small number of observations across time, which limits the application of more sophisticated time-series models. The odds ratio (OR) was used as the measure of association. A significance level of α = 0.05 was adopted, corresponding to a 95% confidence interval. To avoid issues related to zero event counts in stratified analyses, adjacent age groups were combined (e.g., 40–59 years) to ensure that odds ratios could be calculated using valid data from both populations.

All statistical analyses were performed using Microsoft Excel^®^ and the R statistical computing environment (R Development Core Team), version 3.5.

## 3. Results

During the analyzed period, approximately 25,000 suicide deaths were recorded in the southern region of Brazil, of which 68 occurred among the Indigenous population, representing 0.27% of deaths. This proportion is similar to the share of individuals who self-identify as Indigenous in the region, which is 0.28% (Table 1).

The age-standardized suicide mortality rate over the decade was 89.15 per 100,000 inhabitants for the Indigenous population and 92.90 for the non-Indigenous population. Paraná was the only state where a comparatively higher rate was observed among the Indigenous population: 124.14 versus 69.43 per 100,000 inhabitants. Of all suicide deaths among Indigenous people in the southern region, 47% occurred in Paraná.

Males were more affected in both populations and across all states. Paraná exhibited the highest suicide rates among Indigenous individuals, regardless of sex (Figure 1). The odds ratios between the populations were significant: suicide was almost twice as likely among Indigenous males (OR = 1.69; 95% CI: 1.13–2.53) and females (OR = 2.25; 95% CI: 1.13–4.52) (Table 2).

When analyzing age, Indigenous individuals died by suicide at younger ages, with rates of 5.57 versus 0.93 per 100,000 in the 0–14-year age group and 27.97 versus 4.77 in the 15–19-year age group. This corresponds to a sixfold greater risk among Indigenous children (OR = 6.02; 95% CI: 2.48–14.62). However, the data suggest that being Indigenous above age 40 was a protective factor compared to the general population (Table 2).

Regarding educational attainment, literate Indigenous individuals who had not completed secondary school had the highest suicide rate: 68.48 per 100,000 inhabitants (OR = 1.87; 95% CI: 1.29–2.71). In both populations, the lowest rates were observed among those who had not completed primary school. Regarding marital status, the highest rates for the non-Indigenous population were among separated individuals. No suicide deaths were recorded among divorced Indigenous individuals—a subgroup that represented 5.5% of the total.

In 2010, there were 2150 suicide deaths among non-Indigenous individuals in the southern region, increasing to 3154 in 2019 (Figure 2). Among Indigenous individuals, the absolute number of deaths was small—4 in 2010 and 13 in 2019—representing a 325% increase, compared to a 45% increase among the non-Indigenous population. Notably, Rio Grande do Sul consistently had the highest annual suicide rate when considering the total population.

The Mann–Kendall trend test results indicate statistically significant increases over time (*p* < 0.01) in the non-Indigenous population. In contrast, no statistically significant trend was observed among the Indigenous population (Table 3). However, a curious seasonal pattern was noted in some states, as illustrated in the graphs (Figure 2).

## 4. Discussion

Numerous studies have addressed the phenomenon of suicide among Indigenous populations—whether in Brazil [4], Latin America [10], or other parts of the world [11]. Specifically in Brazil, previous research has analyzed the sociodemographic characteristics of vulnerable populations [12], with a focus on specific regions [13] or ethnic groups [14]. The present study examined suicide among the Indigenous population of southern Brazil, the region with the highest general suicide mortality rate in the country for several decades [15].

Time-series studies are widely used and have consistently shown an upward trend in suicide mortality rates in Brazil [16,17]. Our findings corroborate this pattern in the general population of the southern region, with a clear upward trend. It is noteworthy that the amplitude of the values observed among Indigenous individuals was greater, although no statistically significant trend was identified using the applied statistical model. However, it must be considered that the number of events in the Indigenous group was very small (68 cases over 10 years), which may have limited the statistical power. The age-standardized suicide mortality rate among Indigenous people rose from 5.32 per 100,000 inhabitants in 2010 to 17.30 in 2019. This is not unprecedented; rapid increases have historically occurred among other Indigenous populations worldwide. For example, suicide rates increased by 800% among Aboriginal peoples and Torres Strait Islanders in Australia between 1981 and 2002, and by 1600% among the Inuit of Greenland over approximately 100 years [18].

The seasonal pattern observed in suicide mortality among the Indigenous population of southern Brazil is noteworthy, although it was not statistically evaluated in this study. Nonetheless, seasonal suicide patterns are well documented in the literature, typically within a single year, with some countries showing unimodal or bimodal peaks, often with a higher incidence in spring [19]. In this study, the observed pattern was annual, with years of peak incidence followed by marked declines. Although this could be a random finding due to the small number of events, three peaks were observed over a 10-year period. If this pattern proves consistent, two hypotheses may be considered: (1) the Werther effect [20], a phenomenon in which suicide rates increase following a specific and publicized event—also referred to as the copycat phenomenon; and (2) preventive responses, in which health and social support systems respond to sudden increases in suicide deaths by implementing targeted interventions.

Based on data from Indigenous Lands and Reserves (ILs and IRs, respectively) from the 2010 IBGE census [21], it is possible to compare suicide mortality rates by analyzing deaths within municipalities where reserves are located and dividing by the respective populations. For example, in the Avá-Guarani IR (Ocoi), there were 630 inhabitants in 2010, and in the Faixinal IL, there were 605 inhabitants. The respective suicide rates between 2010 and 2019 were 634.92 (4 deaths) and 330.58 (2 deaths) per 100,000 inhabitants [22]. The primary difference between these two communities is ethnicity: Guarani Kaiowá in the former and Kaingang in the latter. A study conducted by Lazzarini et al. (2018) [23] in villages near Dourados (State of Mato Grosso do Sul) examined the phenomenon among youth. In Bororo village, the suicide rate was 115 per 100,000 inhabitants. In Jaguapiru, where the Guarani population constitutes less than 30% of the Indigenous population, the probability of suicide was five times lower [23].

Between 2010 and 2019, 25 suicide deaths (37%) occurred among Indigenous youth aged 15–19. Nationally, Indigenous children aged 10 to 14 have an 18-times higher suicide mortality rate compared to the general population of the same age and location, suggesting localized endemic patterns and intergenerational family histories [24]. Moreover, in cases involving Indigenous youth, there appears to be an intrafamilial pattern, with multiple occurrences at the same address, as suggested by clustering analyses [23]. It is well documented that a positive family history of suicide in a first-degree relative is a major risk factor for planned suicide attempts [25].

In response to the high prevalence of suicide among Indigenous youth, particularly in contexts marked by intergenerational patterns and social vulnerability, some communities have developed culturally tailored prevention strategies. A notable example is the initiative conducted in Naujaat, Canada, where Inuit adolescents were engaged through a participatory, community-based approach to suicide prevention [26]. The program emphasized cultural identity, intergenerational dialogue, and youth protagonism, which collectively contributed to strengthening protective factors such as social cohesion and resilience.

Regarding marital status, the highest prevalence of suicide deaths was among single Indigenous individuals, accounting for approximately 65% of cases in the study region. This likely overlaps with the age distribution, as most single individuals were younger. The “marriage protection” theory is well established. This protective factor appears stronger among men and young individuals and is not limited to suicide as a cause of death [27].

The limitations of this study stem primarily from outdated demographic data. The Brazilian census was conducted in 2010 [5], and it remains the most reliable source of population information. For this reason, alternative population estimation methods, such as interpolation between two known points, could not be applied. This limitation was addressed through estimation techniques consistent with those used in official studies [8]. Although widely accepted, this approach may not fully capture demographic dynamics specific to Indigenous populations, introducing some uncertainty into rate estimates, particularly in trend analyses. Furthermore, we used the direct method of age standardization, as recommended by the World Health Organization for population comparisons; we recognize that small counts in some age groups within the Indigenous population may have introduced statistical instability into the resulting rates.

Another relevant limitation is the geographic dispersion of the target population. Approximately half of the Indigenous population in the studied region lives in Indigenous lands [28], while the other half resides in urban centers. However, the results reflect cumulative events regardless of this distinction. In Brazil, all deaths resulting from external causes or suspected violent events, such as suicide, are required by law to be certified by the Medical Examiner’s Office (Instituto Médico Legal—IML). However, IML units are not available in all cities or rural areas, including Indigenous territories. In some cases, by the time the IML is contacted, funeral rituals may have already taken place. In such situations, two complementary forms may be used: (1) the “Complementary Form for the Notification and Investigation of Suicide Attempts and Deaths among Indigenous Peoples” and (2) the “Death Investigation Form for Ill-Defined Causes” (IOCMD/SVS) [28]. Although these tools aim to improve data quality in vulnerable populations, their application is not always consistent. These systemic limitations may introduce information bias into mortality databases and should be considered when interpreting the findings. Moreover, this study did not include any field verification of death records due to a lack of authorization and operational constraints.

It is also important to note that the apparent absence of data in some categories reflects the lack of recorded events rather than missing information in the dataset. However, in the context of suicide reporting—especially in rural and Indigenous communities—underreporting remains a serious concern. Several structural factors may contribute to this, including the absence of Medical Examiner Offices or Death Verification Services (SVOs) in remote areas, the refusal of families to authorize autopsies, inaccuracies in the completion of death certificates by non-specialist physicians, and technical inconsistencies during data entry into the national Mortality Information System (SIM). These limitations may disproportionately affect Indigenous populations living in areas with limited access to forensic or investigative services, thus introducing potential bias in the detection and classification of suicide deaths.

In summary, the greatest risk of suicide death was observed among Indigenous individuals who were male, young, and single, particularly those residing in the state of Paraná.

## 5. Conclusions

This study aimed to provide epidemiological data to improve the understanding of the suicide phenomenon in vulnerable populations. Indigenous peoples are particularly affected, highlighting the need for interventions tailored to the specific characteristics of each region and ethnic group.

The findings are partially consistent with studies from other regions of Brazil that report elevated suicide rates among Indigenous populations. In our analysis, the overall suicide mortality rate in the southern region did not differ substantially between Indigenous and non-Indigenous groups. However, the state of Paraná showed a notably higher suicide rate among Indigenous individuals, particularly in younger age groups. This pattern aligns with national trends observed in previous studies, which have highlighted increasing suicide rates among Indigenous youth and variability across ethnicities. The issue remains particularly concerning, as it disproportionately affects a segment of the population that is expected to be socially and biologically productive, potentially contributing to demographic decline within these communities.

Future studies should address the cultural understanding of psychiatric conditions and their epidemiology, as these are significant risk factors and crucial targets for prevention strategies. In addition, the use of longitudinal designs may help identify dynamic patterns and long-term outcomes related to suicide risk. Qualitative and mixed-method approaches are also essential to explore sociocultural meanings, family and community narratives, and subjective experiences that cannot be captured by ecological designs alone. These methodologies would provide a more comprehensive understanding of suicide within Indigenous contexts and support the development of more tailored and effective interventions.

Finally, it is important to emphasize that good scientific, medical, and psychosocial practices are essential for recognizing, preventing, and developing responses to this problem—efforts that must overcome not only cultural and linguistic barriers but also a history of conflict, which has generated distrust and segregation.

## Figures and Tables

**Figure 1 ijerph-22-01313-f001:**
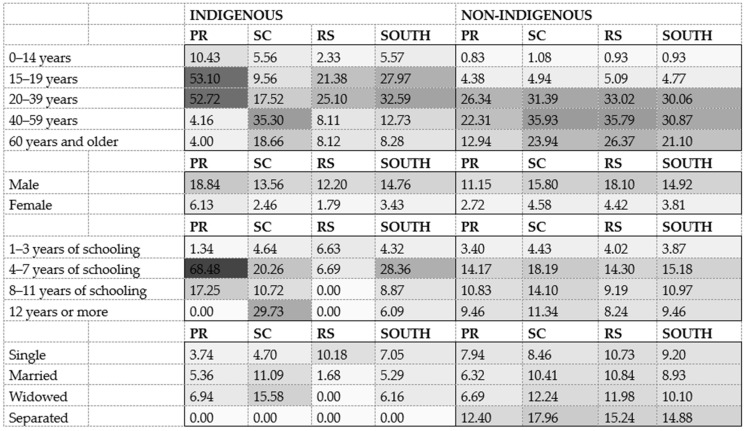
Standardized suicide rates by state, age group, sex, and population group in southern Brazil. Source: Authors’ elaboration based on SIM/DATASUS data.

**Figure 2 ijerph-22-01313-f002:**
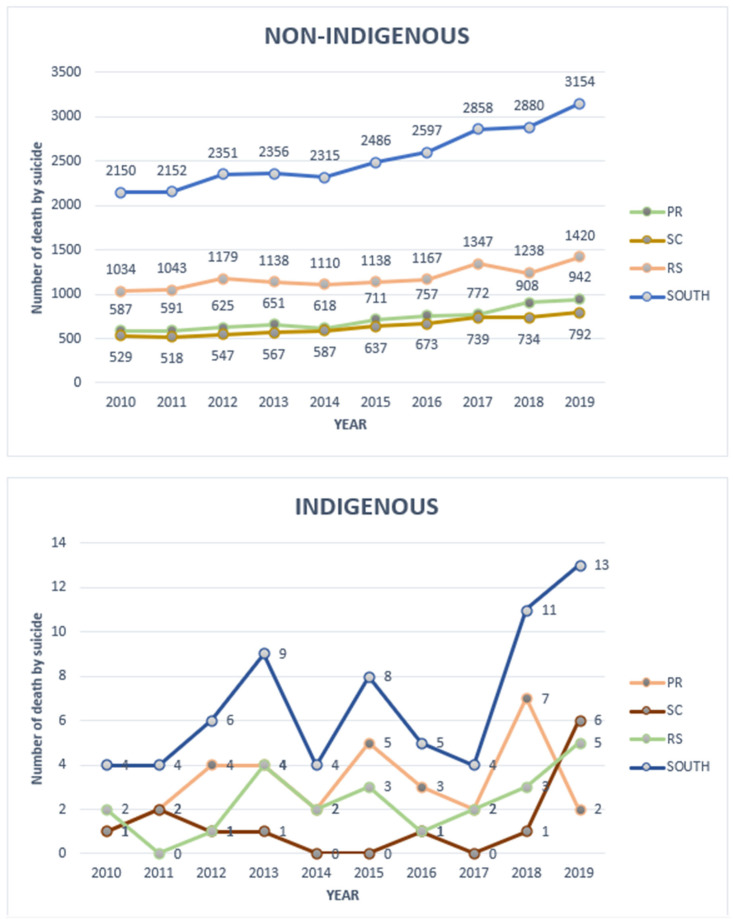
Annual number of suicide deaths by state and population groups in southern Brazil in 2010–2019. Source: Authors’ elaboration based on SIM/DATASUS data.

**Table 1 ijerph-22-01313-t001:** Frequency distribution of sociodemographic characteristics by population and state in southern region of Brazil.

Variable	State						South Region	
	PR		SC		RS			
	Indigenous	Non-Indigenous	Indigenous	Non-Indigenous	Indigenous	Non-Indigenous	Indigenous	Non-Indigenous
Population in 2010	25,786	10,418.741	16,242	6,232,194	33,152	10,660,778	75,180	27,311,713
Sex								
Female	8 (25%)	1451 (20.18%)	2 (15.38%)	1440 (22.73%)	3 (13.04%)	2424 (20.48%)	13 (19.12%)	5315 (20.95%)
Male	24 (75%)	5741 (79.82%)	11 (84.62%)	4896 (77.27%)	20 (86.96%)	9413 (79.52%)	55 (80.08%)	20,050 (79.05%)
Age								
5–9 years	0 (0%)	1 (0.01%)	0 (0.00%)	1 (0.02%)	1 (4.55%)	2 (0.02%)	1 (1.49%)	4 (0.02%)
10–14 years	3 (9.38%)	78 (1.07%)	1 (7.69%)	56 (0.88%)	0 (0.00%)	78 (0.66%)	4 (5.97%)	212 (0.83%)
15–19 years	14 (43.75%)	493 (6.79%)	2 (15.38%)	319 (5.03%)	9 (40.91%)	533 (4.51%)	25 (37.31%)	1345 (5.29%)
20–29 years	9 (28.13%)	1445 (19.89%)	2 (15.38%)	1031 (16.27%)	6 (27.27%)	1650 (13.95%)	17 (25.37%)	4126 (16.22%)
30–39 years	4 (12.50%)	1463 (20.14%)	1 (7.69%)	1114 (17.58%)	2 (9.09%)	1891 (15.98%)	7 (10.45%)	4468 (17.57%)
40–49 years	0 (0.00%)	1422 (19.58%)	5 (38.46%)	1242 (19.60%)	2 (9.09%)	2171 (18.35%)	7 (10.45%)	4835 (19.01%)
50–59 years	1 (3.13%)	1095 (15.07%)	0 (0.00%)	1259 (19.87%)	0 (0.00%)	2285 (19.31%)	1 (1.49%)	4639 (18.24%)
60–69 years	0 (0.00%)	748 (10.30%)	1 (7.69%)	746 (11.77%)	1 (4.55%)	1668 (14.10%)	2 (2.99%)	3162 (12.43%)
70 years and older	1 (3.13%)	519 (7.14%)	1 (7.69%)	568 (8.96%)	1 (4.55%)	1553 (13.13%)	3 (4.48%)	2640 (10.38%)
Educational attainment								
None	0 (0%)	265 (3.68%)	1 (7.69%)	130 (2.05%)	1 (4.35%)	275 (2.32%)	2 (2.94%)	670 (2.64%)
1–3 years	2 (6.25%)	1215 (16.89%)	3 (23.08%)	960 (15.15%)	11 (47.83%)	1537 (12.98%)	16 (23.53%)	3712 (14.63%)
4–7 years	20 (62.5%)	2315 (32.18%)	5 (38.46%)	1918 (30.27%)	3 (13.04%)	2526 (21.34%)	28 (41.18%)	6759 (26.64%)
8–11 years	4 (12.5%)	2226 (30.94%)	2 (15.38%)	1891 (29.85%)	0 (0%)	2033 (17.17%)	6 (8.82%)	6150 (24.24%)
12 years or more	0 (0%)	822 (11.43%)	1 (7.69%)	595 (9.39%)	0(0%)	665 (5.62%)	1 (1.47%)	2082 (8.21%)
Not reported	6 (18.75%)	351 (4.88%)	1 (7.69%)	842 (13.29%)	8 (34.78%)	4801 (40.56%)	15 (22.06%)	5994 (23.63%)
Marital status								
Single	22 (68.75%)	3428 (47.65%)	4 (30.77%)	2230 (35.2%)	18 (78.26%)	5140 (43.42%)	44 (64.71%)	10,798 (42.57%)
Married	1 (3.12%)	2332 (32.42%)	4 (30.77%)	2265 (35.75%)	1 (4.35%)	3650 (30.84%)	6 (8.82%)	8247 (32.51%)
Divorced	0 (0%)	597 (8.3%)	0 (0%)	572 (9.03%)	0(0%)	892 (7.54%)	0 (0%)	2061 (8.12%)
Widowed	0 (0%)	324 (4.5%)	1 (7.69%)	329 (5.19%)	0(0%)	684 (5.78%)	1 (1.47%)	1337 (5.27%)
Not reported	5 (15.62%)	190 (2.64%)	1 (7.69%)	432 (6.82%)	3 (13.04%)	1209 (10.21%)	9 (13.24%)	1831 (7.22%)
Other	4 (12.5%)	323 (4.49%)	3 (23.08%)	508 (8.02%)	1 (4.35%)	262 (2.21%)	8 (11.76%)	1093 (4.31%)
ICD-10 code								
Not reported	0(0%)	474 (6.59%)	0 (0%)	413 (6.52%)	3 (13.04%)	637 (5.38%)	3 (4.41%)	1524 (6.01%)
Other	1 (3.12%)	1806 (25.10%)	4 (30.77%)	1227 (19.37%)	3 (13.04%)	2712 (22.91%)	8 (11.76%)	5745 (22.65%)
X70	31 (96.88%)	4914 (68.31%)	9 (69.23%)	4696 (74.12%)	17 (73.91%)	8488 (71.71%)	57 (83.82%)	18,098 (71.34%)

Note: The table presents the distribution of suicide deaths by sex, age group, education level, and marital status, comparing Indigenous and non-Indigenous populations across the states of Paraná (PR), Santa Catarina (SC), and Rio Grande do Sul (RS). The data are expressed in absolute numbers and percentages.

**Table 2 ijerph-22-01313-t002:** Odds ratios comparing suicide risk between Indigenous and non-Indigenous populations by age, sex, education, and marital status in southern Brazil.

Variable		State			South
		PR	SC	RS	Region
SEX	MALE	1.69 (1.13–2.53) *	0.86 (0.47–1.55)	0.67 (0.43–1.04)	0.99 (0.76–1.29)
	FEMALE	2.25 (1.13–4.52) *	0.54 (0.13–2.15)	0.40 (0.13–1.16)	0.90 (0.52–1.55)
AGE	0–14 years	12.51 (3.95–39.66) *	5.16 (0.71–37.27)	2.50 (0.35–17.96)	6.02 (2.48–14.62) *
	15–19 years	13.19 (7.15–20.77) *	1.94 (0.48–7.78)	4.21 (2.18–8.14) *	5.87 (3.95–8.73) *
	20–39 years	2.00 (1.16–3.46) *	0.56 (0.18–1.73)	0.76 (0.38–1.52)	1.08 (0.73–1.62)
	40–59 years	0.19 (0.03–1.32)	0.98 (0.41–2.36)	0.23 (0.06–0.91) *	0.41 (0.21–0.82) *
	60 years and older	0.31 (0.04–2.20)	0.78 (0.19–3.12)	0.31 (0.08–1.23)	0.39 (0.16–0.94) *
EDUCATIONAL ATTAINMENT	1–3 years of schooling	0.39 (0.10–1.58)	1.05 (0.39–2.80)	1.65 (0.93–2.91)	1.12 (0.70–1.77)
	4–7 years of schooling	4.86 (3.12–7.56) *	1.11 (0.46–2.68)	0.47 (0.15–1.45)	1.87 (1.29–2.71) *
	8–11 years of schooling	1.59 (0.60–4.26)	0.76 (0.19–3.04)	-	0.81 (0.36–1.80)
	12 years or more	-	2.63 (0.37–18.73)	-	0.64 (0.09–4.58)
	Not reported	15.66 (6.68–36.70) *	0.22 (0.03–1.60)	0.49 (0.24–0.99) *	0.87 (0.52–1.46)
MARITAL STATUS	Single	0.47 (0.18–1.26)	0.56 (0.21–1.48)	0.95 (0.60–1.51)	0.77 (0.52–1.13)
	Married	0.85 (0.32–2.26)	1.07 (0.40–2.84)	0.16 (0.02–1.10)	0.59 (0.31–1.14)
	Widowed	1.04 (0.15–7.39)	1.27 (0.18–9.09)	-	0.61 (0.15–2.44)
	Divorced	-	-	-	-

Note: The table shows crude odds ratios (ORs) with 95% confidence intervals (CIs) for various sociodemographic variables. An OR > 1 indicates a higher suicide risk among Indigenous individuals. Significant results are marked with an asterisk (*p* < 0.05). Blank cells indicate that no suicide events were recorded in that category during the study period, preventing the calculation of odds ratios.

**Table 3 ijerph-22-01313-t003:** Results of Mann–Kendall trend tests for suicide rates by state and population group in southern Brazil in 2010–2019.

State	Indigenous		Non-Indigenous	
	Tau	*p* Value	Tau	*p* Value
PR	0.105	0.769	0.956	<0.001 *
RS	0.424	0.120	0.778	0.002 *
SC	0.050	0.925	0.956	<0.001 *
SOUTH	0.556	0.038	0.944	0.002 *

Note: Tau values represent the correlation coefficient of trend direction. A statistically significant trend (*p* < 0.05) is marked with an asterisk.

## Data Availability

The data used in this study are publicly available through official government platforms, including the Mortality Information System (SIM) and the Brazilian Institute of Geography and Statistics (IBGE). The processed datasets generated during the study are available from the corresponding author upon reasonable request.

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
