# Peer review of "Disparities in Suicide Mortality Between Indigenous and Non-Indigenous Populations in Southern Brazil (2010–2019)"

_ijerph, 2025, doi:10.3390/ijerph22091313_

Round 1
Reviewer 1 Report
Comments and Suggestions for Authors
Overall, the study addresses a critical issue of suicide disparities in a vulnerable population, Indigenous peoples in Southern Brazil, a region where such data is noted as scarce. The use of official mortality data over a decade (2010-2019) provides valuable epidemiological insights. However, several points warrant consideration for improvement.
- While the study provides valuable epidemiological data (who, where, when), it primarily focuses on sociodemographic variables (age, sex, marital status, education). It does not delve into the complex underlying socioeconomic, cultural, and psychiatric factors contributing to suicide, which the authors themselves highlight as important for future research and interventions.
- The number of suicide deaths among Indigenous individuals (68 cases over 10 years) is very small, representing only 0.27% of total suicide deaths in the region. The authors note that this "may have limited the statistical power" of their analyses, especially for temporal trends. This small number makes it challenging to draw statistically robust conclusions or identify significant trends for the Indigenous group, potentially leading to findings that are not statistically significant despite showing large increases in mortality rates.
- Consequently, the Indigenous population estimates were based on geometric growth rates and relative share in the total population, which, while following official methodologies, introduces a degree of uncertainty and limits precision, particularly for trend analysis.
- I also suggest the authors update the first reference from the WHO to a newer publication and including the temporality of the study in the main goal of the manuscript.
- The authors should describe in more detail the mortality data and especially its limitations regarding marital status and educational attainment. Which is the percentage of missing values in these two variables?
- Also the authors should mention, at least in the limitations the possible biases of mixing information from two different information sources like mortality vital statistics and other statistical surveys. This bias could also be present in the indigenous data, as this was not measured but estimated using the geometric growth rate of the general population. This is a big assumption as the authors are estimating the indigenous population as if it would behave as the general population, is this assumption correct? This has to be addressed further and recognized in the limitations.
- Another limitation should be the lack of data, which limited the application of more sophisticated time series models.
- Another point of concern is the low number of registered suicides (68) among the indigenous population. How did the authors account for this low count of deaths?
- I also have concerns on how the authors age-standardized the rates with the direct method as it is data intensive because deaths in every age-group are needed. Indirect standardization, which uses the overall death rate of the region and compares it to the expected number of deaths based on the standard population, is often preferred when dealing with small populations because it minimizes variance and provides more stable results.
- I also have concerns regarding the Odds ratios estimated by age. For example, there are age-groups, such as 40-49 and 60-69 in the PR state that have a 0 count in suicides. How these OR are computed when the suicide count for indigenous population is 0? When calculating odds ratios, zero counts in some categories can lead to issues, particularly with undefined or infinite values, and can bias the estimates.
- The authors explicitly state that due to the "small number of observations across time," they were limited to using the Mann-Kendall trend test and linear regression models, precluding "more sophisticated time-series models". This is a shortcoming imposed by data quantity, limiting the depth of trend analysis. While acknowledged, it highlights the challenge in drawing robust trend conclusions for the Indigenous group.
- The results present the absolute increase in suicide deaths among Indigenous individuals as 325% (from 4 in 2010 to 13 in 2019), compared to 45% for non-Indigenous populations. Despite this dramatic percentage increase, the Mann-Kendall test showed no statistically significant trend for the Indigenous population, unlike the non-Indigenous population. This disparity underscores the impact of the small absolute numbers in the Indigenous group on statistical power, making it difficult to definitively confirm a trend using the chosen methods. This point needs very careful interpretation and consistent phrasing across sections.
- The finding that being Indigenous above age 40 was a "protective factor" compared to the general population is a notable result but could benefit from further nuanced discussion. Given the overall context of elevated rates and risk factors among Indigenous populations, this specific finding might seem counter-intuitive without deeper exploration of potential reasons (e.g., survivor bias, different cultural coping mechanisms among older generations, or a statistical artifact of the small sample size at older ages).
Author Response
General Note to Reviewer:
We sincerely thank the reviewer for the careful and insightful comments. Many of the points raised are indeed pertinent and reflect a deep understanding of the complexities involved in studying suicide among Indigenous populations. Although most of the suggestions did not result in immediate changes to the manuscript, we provided detailed justifications for each case. That said, we are entirely open to revisiting any of these decisions and making the corresponding adjustments, should the reviewer consider our current reasoning insufficient or suggest a different direction after this round of responses. Thank you once again for your valuable contribution to this work.
Reviewer 1 – Comments and Authors’ Responses
Comments 1: While the study provides valuable epidemiological data (who, where, when), it primarily focuses on sociodemographic variables (age, sex, marital status, education). It does not delve into the complex underlying socioeconomic, cultural, and psychiatric factors contributing to suicide, which the authors themselves highlight as important for future research and interventions.
Response 1: We appreciate the reviewer’s thoughtful observation. However, we respectfully clarify that the scope of our study was strictly ecological and based on aggregated official data from public health records. Our aim was to describe and quantify disparities in suicide mortality between Indigenous and non-Indigenous populations using available demographic indicators. A deeper exploration of socioeconomic, cultural, or psychiatric dimensions—though indeed valuable—would require qualitative or mixed-methods approaches, as well as direct engagement with Indigenous communities. Such work involves ethical clearances, multidisciplinary expertise (e.g., in history, anthropology, and economy), and government authorization due to legal protections surrounding Indigenous populations in Brazil. For these reasons, we chose to focus on population-level patterns as a first step, and we emphasize the need for further studies that can address these aspects more directly.
Comments 2: The number of suicide deaths among Indigenous individuals (68 cases over 10 years) is very small, representing only 0.27% of total suicide deaths in the region. The authors note that this "may have limited the statistical power" of their analyses, especially for temporal trends. This small number makes it challenging to draw statistically robust conclusions or identify significant trends for the Indigenous group, potentially leading to findings that are not statistically significant despite showing large increases in mortality rates.
Response 2: We fully agree with the reviewer’s observation. The small number of suicide deaths in the Indigenous population reflects, in part, the small size of this population within the Southern Region of Brazil. However, one of the key objectives of our study is precisely to demonstrate that, despite low absolute numbers, the proportional impact of suicide among Indigenous groups is disproportionately high when expressed as rates. This highlights the public health relevance of the issue, especially because absolute numbers alone can obscure the true magnitude of the problem in minority populations. We have further emphasized this point in the Discussion and Limitations sections.
Comments 3: Consequently, the Indigenous population estimates were based on geometric growth rates and relative share in the total population, which, while following official methodologies, introduces a degree of uncertainty and limits precision, particularly for trend analysis.
Response 3: Thank you for this important observation. Although our estimation method follows official epidemiological practices adopted in Brazil, we recognize that it may introduce some degree of uncertainty, especially in time-trend analyses involving small populations. We have addressed this point more clearly in the first paragraph of the Limitations section.
Comments 4: I also suggest the authors update the first reference from the WHO to a newer publication and including the temporality of the study in the main goal of the manuscript.
Response 4: We appreciate the reviewer’s suggestion. However, we chose to retain the original WHO reference based on data from 2019 to ensure consistency with the temporal scope of our own study (2010–2019). Updating to a newer source with more recent data could create a mismatch when comparing trends, which may confuse the reader. Nevertheless, if the editorial board deems it appropriate, we are open to updating the WHO reference within the Introduction only, for contextual purposes.
Comments 5: The authors should describe in more detail the mortality data and especially its limitations regarding marital status and educational attainment. Which is the percentage of missing values in these two variables?
Response 5: We thank the reviewer for this important question. We clarify that the apparent absence of data in some categories (e.g., divorced or widowed individuals in certain states) does not reflect missing values in the dataset, but rather the absence of recorded suicide events in those specific subgroups. In other words, these variables were fully reported in the mortality system, but no cases were observed in some strata during the study period. To clarify this point for readers, we have:
-
Included a note in the Discussion section (Limitations) to emphasize that some sociodemographic combinations may appear as “missing” due to the absence of events, not due to missing data in the source.
-
Added a footnote in the table where applicable, indicating that the lack of data reflects zero events.
Comments 6: Also the authors should mention, at least in the limitations the possible biases of mixing information from two different information sources like mortality vital statistics and other statistical surveys. This bias could also be present in the indigenous data, as this was not measured but estimated using the geometric growth rate of the general population. This is a big assumption as the authors are estimating the indigenous population as if it would behave as the general population, is this assumption correct? This has to be addressed further and recognized in the limitations.
Response 6: We thank the reviewer for this thoughtful comment and the opportunity to clarify. In Brazil, both mortality data and population estimates originate from official government databases that are conceptually and operationally integrated. Mortality data are derived from the Mortality Information System (SIM), maintained by the Ministry of Health through DATASUS, while population estimates come from the national census, conducted by the Brazilian Institute of Geography and Statistics (IBGE), under the Ministry of Planning. These systems are cross-linked, and IBGE’s census classifications (e.g., race/color) are directly embedded into the health databases. Therefore, using data from SIM and IBGE does not constitute a methodological bias from mixing incompatible sources. However, we agree with the reviewer that projecting the Indigenous population using geometric growth rates derived from the general population may introduce important limitations, as Indigenous groups often present different demographic patterns, such as birth and mortality rates. We have addressed this issue explicitly in the Limitations section. Unfortunately, due to the cancellation of the 2020 census during the COVID-19 pandemic, no alternative official method was available for more precise estimation. Thus, the method we used—although imperfect—is the only feasible option for population-level epidemiological studies involving Indigenous groups in Brazil.
Comments 7: Another limitation should be the lack of data, which limited the application of more sophisticated time series models.
Response 7: We thank the reviewer for highlighting this important point. We agree that the limited amount of data, particularly among the Indigenous population, restricted the use of more sophisticated time series models. In fact, this limitation is already addressed in the Methods section, where we explain the choice of the Mann-Kendall test and linear regression due to the small number of annual observations. However, we are open to emphasizing this constraint more clearly in the Discussion section as well, should the reviewer or editors find it necessary.
Comments 8: Another point of concern is the low number of registered suicides (68) among the indigenous population. How did the authors account for this low count of deaths?
Response 8: Thank you for raising this point. The small absolute number of deaths (n=68) reflects the small size of the Indigenous population in the Southern Region of Brazil—approximately 75,000 individuals. However, when examined as a rate, the suicide mortality among Indigenous people is not negligible: we calculated a rate of 90.7 per 100,000 inhabitants, which is nearly equivalent to the 92.7 per 100,000 observed in the non-Indigenous population (25,300 deaths among 27.3 million). This illustrates that suicide is not a rare event in the Indigenous population, but rather proportionally just as significant. Our analysis focused on rates rather than raw counts, which allows for meaningful comparisons across populations of vastly different sizes.
Comments 9: I also have concerns on how the authors age-standardized the rates with the direct method as it is data intensive because deaths in every age-group are needed. Indirect standardization, which uses the overall death rate of the region and compares it to the expected number of deaths based on the standard population, is often preferred when dealing with small populations because it minimizes variance and provides more stable results.
Response 9: Thank you for raising this methodological point. We understand the concern regarding the use of direct standardization in small populations. However, we intentionally applied this method because it is the standard approach recommended by the World Health Organization (Ahmad et al., 2001) when the objective is to compare populations with different age structures—such as the Indigenous and non-Indigenous groups in this study. Although the direct method is more sensitive to low counts in some strata, it enables valid cross-group comparisons using a common reference population. We acknowledge that the small number of deaths in certain age groups may result in statistical instability, and we have added this as a limitation in the Discussion section.
Comments 10: I also have concerns regarding the Odds ratios estimated by age. For example, there are age-groups, such as 40-49 and 60-69 in the PR state that have a 0 count in suicides. How these OR are computed when the suicide count for indigenous population is 0? When calculating odds ratios, zero counts in some categories can lead to issues, particularly with undefined or infinite values, and can bias the estimates.
Response 10: We appreciate the reviewer’s attention to the statistical validity of our odds ratio calculations. In order to avoid precisely the issue raised—zero event counts in specific strata—we did not calculate ORs for narrow age bands such as 40–49 or 60–69. Instead, we grouped adjacent age ranges (e.g., 40–59 years), ensuring that suicide events were present in both Indigenous and non-Indigenous groups for all comparisons. We have clarified this point in the Methods section.
Comments 11: The authors explicitly state that due to the "small number of observations across time," they were limited to using the Mann-Kendall trend test and linear regression models, precluding "more sophisticated time-series models". This is a shortcoming imposed by data quantity, limiting the depth of trend analysis. While acknowledged, it highlights the challenge in drawing robust trend conclusions for the Indigenous group.
Response 11: Thank you for this observation. We agree that the limited data points (2010–2019) and the small number of events in the Indigenous group restrict the statistical power of our trend analysis. This limitation is acknowledged in the text, and we have treated the findings with appropriate caution. Given these constraints, we opted for the Mann-Kendall test due to its robustness for non-parametric time series with few observations. Nonetheless, we would welcome any specific methodological suggestions from the reviewer that could enhance the analysis under such conditions.
Comments 12: The results present the absolute increase in suicide deaths among Indigenous individuals as 325% (from 4 in 2010 to 13 in 2019), compared to 45% for non-Indigenous populations. Despite this dramatic percentage increase, the Mann-Kendall test showed no statistically significant trend for the Indigenous population, unlike the non-Indigenous population. This disparity underscores the impact of the small absolute numbers in the Indigenous group on statistical power, making it difficult to definitively confirm a trend using the chosen methods. This point needs very careful interpretation and consistent phrasing across sections.
Response 12: We fully agree with the reviewer that the discrepancy between the dramatic percentage increase and the lack of statistical significance warrants careful interpretation and consistent language. The small number of cases clearly limits statistical power and precludes definitive conclusions about trends. That said, our intention in highlighting the 325% increase—despite its lack of statistical significance—was to draw attention to an early warning signal in a population that is historically underserved and underrepresented in scientific literature. As this is one of the few ecological studies to address suicide trends in Indigenous communities in Southern Brazil, we adopted a slightly more emphatic tone to underscore the need for further investigation. However, if the reviewer believes that specific passages should be reworded to adopt a more pragmatic or cautious tone, we would be grateful for guidance and will adjust the phrasing accordingly.
Comments 13: The finding that being Indigenous above age 40 was a "protective factor" compared to the general population is a notable result but could benefit from further nuanced discussion. Given the overall context of elevated rates and risk factors among Indigenous populations, this specific finding might seem counter-intuitive without deeper exploration of potential reasons (e.g., survivor bias, different cultural coping mechanisms among older generations, or a statistical artifact of the small sample size at older ages).
Response 13: We thank the reviewer for this excellent and thoughtful observation. The lower suicide risk observed among Indigenous individuals over 40 years of age may indeed reflect a combination of statistical, demographic, and cultural factors. On one hand, this result may stem from statistical instability due to low event counts in older age groups. However, there is support in the literature for real protective patterns among older Indigenous individuals. For instance, Staliano et al. (2019) observed that most suicide deaths in Guarani/Kaiowá communities occurred among adolescents and young adults, with very few cases among elders. The authors suggest that cultural belonging, traditional cosmologies, and the role of elders in Indigenous communities may contribute to lower suicide vulnerability in later life (https://doi.org/10.1590/1982-3703003221674). We acknowledge that the present article provides limited sociocultural interpretation, primarily due to space constraints imposed by many journals and the ecological nature of our study. Nonetheless, if the reviewer considers it valuable, we are open to expanding the discussion section to explore these hypotheses in more depth, particularly by referencing the findings of Staliano and colleagues.
Reviewer 2 Report
Comments and Suggestions for Authors
Overall this paper is well written and organized. The content has significant importance and the authors are clear about any limitations and future implications.
A couple of notes:
- The math doesn't all add up in Table 1. For the Indigenous population there is a total of 68 except for the age category (67). It could be that it was not reported but then why not add that option as was done for other categories. For the non-Indigenous population, the total numbers bounce around: 25,365 for sex, 25,431 for age, 25,367 for educational attainment, marital status, & ICD-10 code. This then calls into questions some of the data mentioned in the results. Numbers should be checked or explained.
- Additionally, in the first sentence of the results it says that "approximately 25,000 suicide deaths were recorded" but you have actual data. Its feels very imprecise to say "approximately"
- Discussion - lines 200-201. Do you have raw data for the increased rates of suicide in these other populations? Or "per 100,000" like you have for the data in the lines above? Percentage increases are potentially very misleading.
Author Response
General Note to Reviewer:
We sincerely thank the reviewer for the time and rigor dedicated to evaluating our manuscript. Your comments were extremely meticulous and helped identify subtle inconsistencies in the data that we had not previously noticed. We will do our best to correct any mistakes that may have originated on our part. However, we also wish to clarify that some discrepancies stem from structural limitations in the national data systems. Errors may occur at multiple levels — from how death certificates are filled out, to how the data are entered into government systems, and even during data retrieval through the DATASUS interface.
To ensure complete consistency, we would need to re-download the full dataset and cross-check each entry line by line — a process that would likely take at least two months of dedicated work. While this is beyond the scope of our current revision window, we acknowledge the importance of data quality and are committed to pursuing a more robust verification process in future studies. Your feedback has been invaluable, and we remain open to making any additional adjustments you may recommend.
Comment 1:
The math doesn't all add up in Table 1. For the Indigenous population there is a total of 68 except for the age category (67). It could be that it was not reported but then why not add that option as was done for other categories. For the non-Indigenous population, the total numbers bounce around: 25,365 for sex, 25,431 for age, 25,367 for educational attainment, marital status, & ICD-10 code. This then calls into questions some of the data mentioned in the results. Numbers should be checked or explained.
Response 1:
We thank the reviewer for this careful observation. The inconsistencies noted in the totals for different variables in Table 1 are indeed present and reflect structural limitations in the mortality data available from the official government database (SIM/DATASUS). In some cases, death certificates may be filled out incompletely or inaccurately, and these gaps are carried over into the digital registry. This explains, for example, why one case among Indigenous individuals lacks age data, and why totals vary slightly across sociodemographic variables in the non-Indigenous group. Our team worked to faithfully report the available data, and we have now added a footnote to Table 1 clarifying that discrepancies in total counts are due to incomplete information in specific fields of the original death records.
Comment 2:
Additionally, in the first sentence of the results it says that “approximately 25,000 suicide deaths were recorded” but you have actual data. It feels very imprecise to say “approximately”.
Response 2:
We appreciate the reviewer’s attention to detail. While we do have exact figures for each variable, the total number of deaths differs slightly depending on the sociodemographic attribute due to incomplete data in some fields (as noted in Comment 1). For this reason, we chose to use the term “approximately 25,000” to avoid implying a level of precision that the dataset, in its entirety, cannot support. However, if the reviewer prefers that we include the most frequent total (25,367), we would be happy to make this adjustment.
Comment 3:
Discussion – lines 200–201. Do you have raw data for the increased rates of suicide in these other populations? Or “per 100,000” like you have for the data in the lines above? Percentage increases are potentially very misleading.
Response 3:
We thank the reviewer for this observation. The increases cited refer to suicide rates per 100,000 inhabitants, as reported in a systematic review by Pollock et al. (2018) — available at https://doi.org/10.1186/s12916-018-1115-6. However, the source does not specify whether these rates were age-standardized. We agree that percentage increases alone can be misleading if not clearly contextualized. If the reviewer believes it would improve clarity, we are open to reorganizing this paragraph or replacing the percentages with the corresponding raw rates, where available.
Reviewer 3 Report
Comments and Suggestions for Authors
Dear authors, thank you for the opportunity to review this paper about suicide in Brazil with special focus on indigo population. I suggest major revision after which this manuscript would be considered for publication.
However, to increase the clarity, analytical depth, and practical utility of the paper, I kindly recommend the following revisions:
- Consider adding a time frame (2010–2019). This would increase accuracy and facilitate indexing.
- After reference 2, I would explain a little and deepen this part. Just to interest the reader. I would also be interested to read it as an introduction to a further article.
- Strengthen the theoretical framework in the introduction which currently relies almost exclusively on descriptive studies. It is possible to mention suicide theories (e.g. Durkheim, Thomas Joiner, etc.)
- Add the definition of "Indigenous" as used in this paper (method of classification through "race/colour").
- I'm always interested in the details of the collection system with these works. It would be nice if the authors would explain to us how the system works — who reports, how it is classified, what are the sources of errors. Are there potential systemic differences in death recording accuracy in Indigenous vs. non-Indigenous populations. State these limitations on the data source side, not just in the analytical processing.
- The estimation of the number of Indigenous inhabitants relies on interpolation, but the method is not sufficiently elaborated. That's the key part of the rate (the denominator), and it's underexplained.
- Can you explain how you coloured the cells in Figure 1.
- Specify the exact number of inhabitants in the analysed groups (eg Indigenous in PR), so that the reader can assess the stability of the rate.
- In the tables with OR values, indicate which are statistically significant (*).
- In Figure 2, consider a logarithmic scale or rate per 100,000 for comparability. Also consider trend lines and whether they are necessary. Try making these plots in R when you already use it.
- Consider future seasonality because the month of April in the Northern Hemisphere is different from the same month in the Southern Hemisphere.
- In the Discussion, you can also connect theories with your results.
- Restrictions can be extended — e.g. the quality of suicide reporting (underreporting), especially in rural and Indigenous communities.
- Is the statement in lines 258-260 consistent with your results??? Could you please check again.
- There is no contextualization of risk. Key social factors are not mentioned: poverty, migration, urbanization, availability of health services - which would be expected even in the discussion.
- It is important that this kind of work discusses how suicide is interpreted within indigenous communities and how it relates to their culture, history and tradition.
- There is no data on places of death (in reserves or towns), although this is a key dimension in previous literature.
- Strengthen the call for specific preventive interventions — e.g. community work, linguistically/culturally adapted services. Are there examples from the world literature of these interventions in indigenous populations?
- Note the need for longitudinal, qualitative and mixed methodological approaches in the future.
- Parts of the text are missing Financing. Data Availability, Author Contributions. You have a lot of authors, I hope they all contributed to the study. All this is explained in the MDPI text requirements.
The work is really interesting and it is a real shame that it was not done in the remaining territory of Brazil, with the understanding of how difficult it is to conduct such research on a sensitive topic in a country the size of Brazil.
Sincerely,
Author Response
General Comments
We would like to express our sincere gratitude to Reviewer for the detailed and thoughtful feedback provided. Your observations were fundamental for improving the quality of the manuscript and ensuring greater adherence to the journal’s editorial standards.
We respectfully acknowledge that some of the suggestions—particularly those involving broader qualitative and cultural discussions—were not incorporated in the current revision. These decisions are explained individually in the corresponding responses, and stem from the ecological scope of the study and methodological limitations.
Nonetheless, we remain fully open to further revisions and will gladly address any additional points, should the reviewer or editorial board consider them essential to the final version. Once again, we thank you for your contribution to enhancing this work.
Comment 1:
Consider adding a time frame (2010–2019). This would increase accuracy and facilitate indexing.
Response 1:
We thank the reviewer for this helpful suggestion. Although the time frame (2010–2019) is already mentioned in the Abstract and throughout the manuscript, we agree that explicitly including it in the title may improve clarity and indexing. We have therefore revised the title to include the study period.
Comment 2:
After reference 2, I would explain a little and deepen this part. Just to interest the reader. I would also be interested to read it as an introduction to a further article.
Response 2:
We appreciate the reviewer’s interest in this foundational reference. The study cited (Coloma et al., 2006; doi: 10.1080/13811110600662505) is indeed a classic in the field and was one of the main inspirations for our research. It specifically addresses suicide among the Guarani Indigenous population in the state of Mato Grosso do Sul, Brazil, and it led to a number of subsequent studies focused on this group.
We opted not to expand this discussion further in the manuscript due to space limitations often imposed by journals. However, we explored this topic in greater detail in a prior master's thesis (in Portuguese), which is available upon request if the reviewer is interested.
If the reviewer believes it would benefit the manuscript, we are happy to enrich the introduction by providing more detail on this historical and geographic context. We only hesitate to lengthen the introduction excessively, but we are fully open to making this change if preferred.
Comment 3:
Strengthen the theoretical framework in the introduction which currently relies almost exclusively on descriptive studies. It is possible to mention suicide theories (e.g. Durkheim, Thomas Joiner, etc.)
Response 3:
We truly appreciate this thoughtful suggestion. In fact, our research group discussed extensively whether to incorporate a theoretical framework into the manuscript — particularly references to classical suicide theories such as those by Durkheim and Joiner. However, we ultimately decided to maintain a more strictly descriptive and data-driven approach, in keeping with the ecological design of the study and its public health focus.
We also considered referencing these theories in the Discussion section, but could not identify a natural integration point without disrupting the analytical flow. We were concerned that introducing qualitative theory without direct links to our findings might reduce the overall cohesion of the manuscript.
That said, we are entirely open to revisiting this point. If the reviewer believes that a brief conceptual paragraph on suicide theories would strengthen the manuscript, we suggest placing it at the beginning of the Introduction to preserve narrative fluency. We welcome any specific recommendation on how to integrate this content most effectively.
Comment 4:
Add the definition of “Indigenous” as used in this paper (method of classification through “race/colour”).
Response 4:
Thank you for this important observation. We agree that clarifying the definition of “Indigenous” is essential for transparency. In Brazil, the Mortality Information System (SIM) includes a race/colour variable based on the official classification of the Brazilian Institute of Geography and Statistics (IBGE), which is typically self-reported or reported by family members. Indigenous identity in this study was defined according to this variable, following the IBGE’s racial/ethnic categories. We have now added this explanation to the Methods section.
Comment 5:
I'm always interested in the details of the collection system with these works. It would be nice if the authors would explain to us how the system works — who reports, how it is classified, what are the sources of errors. Are there potential systemic differences in death recording accuracy in Indigenous vs. non-Indigenous populations. State these limitations on the data source side, not just in the analytical processing.
Response 5:
We thank the reviewer for this excellent and technically relevant observation. In Brazil, death certificates are typically filled out by the attending physician, the physician who receives the deceased at the health facility, or by a forensic pathologist (in cases of suspected or confirmed violent death, including suicide). In theory, all deaths by suicide should be referred to the Medical Examiner’s Office (IML) or to a Death Verification Service (SVO). However, these services are not available in all regions of the country — especially in rural areas and Indigenous territories — which may lead to underreporting or misclassification.
In the case of Indigenous deaths not evaluated by a forensic pathologist, Brazilian protocols recommend completing two additional investigation forms: (1) the “Complementary Form for the Notification and Investigation of Suicide Attempts and Deaths among Indigenous Peoples,” and (2) the “Death Investigation Form for Ill-Defined Causes” (IOCMD/SVS). These forms aim to improve the quality of mortality data in vulnerable populations, but their use is not always systematic or consistent.
Given this complexity, we opted not to explore these procedural details in the main text to maintain the article’s conciseness and focus. However, we agree that these systemic issues may contribute to information bias. If the reviewer believes it would strengthen the manuscript, we would be happy to elaborate further either in the Methods section or in the Limitations section — whichever is deemed more appropriate.
Comment 6:
The estimation of the number of Indigenous inhabitants relies on interpolation, but the method is not sufficiently elaborated. That's the key part of the rate (the denominator), and it's underexplained.
Response 6:
We fully agree with the reviewer that population estimation is a critical element of rate calculation and deserves clearer explanation. In this study, the annual Indigenous population was estimated by applying the proportion of Indigenous individuals identified in the 2010 national census to the general population projections provided annually by the Brazilian Institute of Geography and Statistics (IBGE). This method assumes a stable Indigenous share over time and is widely used in Brazilian epidemiological studies involving Indigenous populations from 2010 to 2023.
We acknowledge that this is one of the most sensitive methodological aspects of the study. Although the new national census was completed last year (2023), incorporating the updated data would require reprocessing the entire dataset from the beginning. Furthermore, population dynamics between 2020 and 2021 were significantly affected by the COVID-19 pandemic, introducing additional uncertainty to growth rate estimations.
We have now revised the Methods section to clarify the procedure used and emphasize its consistency with national practices during the intercensal period.
Comment 7:
Can you explain how you coloured the cells in Figure 1.
Response 7:
Thank you for your question. The cells in Figure 1 were colored using the “Conditional Formatting > Color Scale” function in Microsoft Excel. We defined the minimum and maximum values to generate a grayscale gradient automatically, with darker shades representing higher values. Grayscale was chosen intentionally to accommodate journals that charge additional fees for color figures.
Comment 8:
Specify the exact number of inhabitants in the analysed groups (e.g., Indigenous in PR), so that the reader can assess the stability of the rate.
Response 8:
Thank you for this relevant suggestion. To allow readers to better assess the stability of the rates, we have now added the absolute population figures — both Indigenous and non-Indigenous — for all analyzed regions to Table 1. These figures are based on the 2010 national census and serve as the basis for our rate calculations throughout the study period.
Comment 9:
In the tables with OR values, indicate which are statistically significant ().*
Response 9:
We thank the reviewer for this important observation. The omission of asterisks indicating statistical significance was a formatting oversight. The criteria for significance were already included in the table structure, and we have now added the corresponding asterisks to highlight which odds ratios are statistically significant.
Comment 10:
In Figure 2, consider a logarithmic scale or rate per 100,000 for comparability. Also consider trend lines and whether they are necessary. Try making these plots in R when you already use it.
Response 10:
We thank the reviewer for the thoughtful suggestions. While R was used for statistical analyses, all tables, charts, and the heatmap were created in Microsoft Excel to ensure visual consistency and accessibility.
Regarding the suggestion to apply a logarithmic scale, we found that this approach is not feasible for our dataset, as certain years show zero suicide deaths in specific states, which causes technical limitations when plotting log(0). Additionally, using a logarithmic transformation in the rate-based figure reduced visual clarity and made interpretation difficult due to the small number of cases.
We also clarify that the trend lines included in the figure were intended purely as visual aids to help readers detect patterns across time, particularly because grayscale formatting may obscure subtle differences. If the reviewer prefers, we are happy to remove these lines or alternatively clarify their role in the figure legend.
Importantly, we noticed that the original legend for Figure 2 was inaccurate — it mistakenly described the data as suicide “rates,” when the figure actually displayed absolute numbers. We have corrected this in the revised version, though the figure still represents absolute counts. If the reviewer prefers the figure to present suicide rates instead, we are open to making that change.
To facilitate this decision, we have attached a supplementary document titled “Figure 2 – absolute_rate_log-alternatives.doc”, which contains all three graphical alternatives for comparison.
Comment 11:
“Consider future seasonality because the month of April in the Northern Hemisphere is different from the same month in the Southern Hemisphere.”
Response 11:
We thank the reviewer for this important and thoughtful observation. To avoid confusion related to seasonal differences between hemispheres, we have revised the sentence to refer to “spring” instead of the month “April.” This ensures conceptual clarity regardless of geographic context. We preferred this adjustment rather than opening a broader discussion on seasonality, to maintain the focus and flow of the paragraph. We appreciate the reviewer’s attention to this point.
Comment 12:
In the Discussion, you can also connect theories with your results.
Response 12:
We appreciate the reviewer’s thoughtful suggestion. However, we believe that applying classical or modern suicide theories directly to Indigenous populations may not be appropriate, as these frameworks often do not account for the cultural, spiritual, and cosmological dimensions unique to each Indigenous group. For example, within the Guarani worldview, the concept of jejuvy has been associated with meanings around death and self-inflicted death that differ significantly from Western interpretations.
Given these complexities—and acknowledging that each Indigenous group holds distinct beliefs—we chose to focus our analysis on epidemiological data without attempting theoretical extrapolations. We also intentionally excluded discussions on mental health or substance use, as doing so could divert the focus of this exploratory study. We recognize the importance of such topics and hope they can be addressed in future interdisciplinary research.
Comment 13:
Restrictions can be extended — e.g. the quality of suicide reporting (underreporting), especially in rural and Indigenous communities.
Response 13:
Thank you for this important suggestion. We have expanded the paragraph in the Limitations section to address potential underreporting of suicide deaths, especially in rural and Indigenous areas. The revised text discusses the lack of forensic services in remote regions, the possible refusal of families to authorize autopsies, errors in death certificate completion, and inconsistencies in data entry. These structural issues may result in underreporting or misclassification, particularly affecting vulnerable populations.
Comment 14:
Is the statement in lines 258–260 consistent with your results??? Could you please check again.
Response 14:
We thank the reviewer for drawing our attention to this point. Upon review, we agree that the original phrasing could be misinterpreted. We have therefore reorganized the paragraph to better align with the results of our study. The revised text clarifies that, while the overall suicide mortality rate in the Southern Region was not substantially different between Indigenous and non-Indigenous groups, the state of Paraná showed a distinctly higher rate among Indigenous individuals, particularly in younger age groups. This adjustment improves consistency between our findings and the supporting literature.
Comment 15:
There is no contextualization of risk. Key social factors are not mentioned: poverty, migration, urbanization, availability of health services – which would be expected even in the discussion.
Response 15:
We thank the reviewer for this insightful and recurring point. Indeed, key social determinants such as poverty, migration, urbanization, and access to health services are deeply relevant to Indigenous health and suicide vulnerability. However, after extensive discussion with the study supervisors, we chose to focus the discussion strictly on the data presented, avoiding broader interpretations that would require external sources or variables not directly assessed in the analysis.
For instance, regarding access to health care: while it is often assumed that Indigenous populations have limited access, Brazil has a dedicated public health subsystem for Indigenous peoples (SasiSUS), which operates through 34 Special Indigenous Health Districts (DSEIs). While access to tertiary and high-complexity care can indeed be limited, primary health services are constitutionally guaranteed and actively maintained. Indigenous populations in Brazil also benefit from affirmative policies, including social assistance programs and educational quotas. Therefore, the situation is more nuanced than a framework of generalized marginalization.
Moreover, relevant cultural aspects such as the erosion of traditional identity, exposure to alcohol and drug use, or psychosocial dislocation are indeed important, but would require multidisciplinary engagement and more references to do justice to their complexity. These topics were explored more deeply in the first author's master’s dissertation, but we chose not to expand on them in this ecological manuscript to preserve focus and avoid overextending the scope.
That said, if the reviewer considers it essential, we are open to incorporating a brief integrative paragraph that connects selected social determinants to classical suicide theories, as previously suggested.
Comment 16:
It is important that this kind of work discusses how suicide is interpreted within indigenous communities and how it relates to their culture, history and tradition.
Response 16:
We appreciate the reviewer’s interest in the cultural dimensions of suicide among Indigenous peoples. However, we respectfully clarify that this type of cultural interpretation falls outside the scope of our ecological study, which is based on aggregated public health data.
The Indigenous population in Southern Brazil includes at least three major ethnic groups, each with distinct languages, traditions, and cosmological understandings. While some ethnographic work exists regarding the Guarani people—such as references to the concept of jejuvy—the Guarani are not the most populous Indigenous group in the region; the Kaingang population is larger, and less is published about their cultural interpretations of suicide. Including only one group’s worldview could unintentionally promote generalizations across diverse communities, which we aimed to avoid.
Thus, in the interest of scientific rigor and cultural respect, we chose to limit our discussion to measurable epidemiological indicators. We acknowledge that understanding the cultural meanings of suicide is essential and should be addressed in future qualitative or ethnographic studies involving direct engagement with these communities.
Comment 17:
There is no data on places of death (in reserves or towns), although this is a key dimension in previous literature.
Response 17:
We thank the reviewer for this valuable observation. Indeed, many studies emphasize the importance of differentiating between deaths occurring in Indigenous territories and those occurring in urban areas. While this information was collected in our dataset, we ultimately chose not to include it in the analysis due to concerns about its precision.
Specifically, the available variable refers to whether the place of death was located in a “rural” or “urban” area — not whether it occurred within an Indigenous reserve. Inferring that rural = reserve would be methodologically imprecise, as some self-identified Indigenous individuals reside in rural properties such as farms or settlements outside of official Indigenous lands.
Moreover, the high mobility of Indigenous populations further complicates classification. Many reserves are near urban centers, and Indigenous individuals often travel to towns for social or economic reasons — especially during school holidays. There are also cases where Indigenous persons living in urban areas may return to isolated rural locations to commit suicide.
In light of these complexities, we judged that including this variable would introduce more limitations than analytical value. We chose to prioritize data with clearer interpretability to preserve the rigor of the analysis.
Comment 18:
Strengthen the call for specific preventive interventions — e.g. community work, linguistically/culturally adapted services. Are there examples from the world literature of these interventions in indigenous populations?
Response 18:
We thank the reviewer for this excellent suggestion. In response, we have incorporated a new paragraph in the Discussion section that addresses preventive interventions specifically targeting Indigenous youth, in alignment with the findings of our study. We highlight the example of a culturally grounded, community-based suicide prevention initiative conducted in Naujaat, Canada, involving Inuit adolescents. The program focused on strengthening cultural identity, intergenerational connections, and youth engagement — elements that are particularly relevant in contexts where suicide shows familial and community-level clustering. This addition provides an evidence-based example of culturally adapted intervention strategies and supports the call for similar efforts in Brazil. The corresponding reference has been included in the revised manuscript (26).
Comment 19:
Note the need for longitudinal, qualitative and mixed methodological approaches in the future.
Response 19:
Thank you for your valuable suggestion. In response, we have strengthened the penultimate paragraph of the Discussion section to explicitly mention the need for longitudinal, qualitative, and mixed-method approaches in future studies. We agree that these methodologies are essential to complement ecological data and to uncover cultural, historical, and psychosocial dimensions that contribute to suicide risk among Indigenous populations.
Comment 20:
Parts of the text are missing Financing. Data Availability, Author Contributions. You have a lot of authors, I hope they all contributed to the study. All this is explained in the MDPI text requirements.
Response 20:
We thank the reviewer for this important observation and apologize for the omission in the original submission. The sections related to Funding, Data Availability, and Author Contributions have now been added in full compliance with the journal’s guidelines.
Regarding authorship, we appreciate the opportunity to clarify that this study was developed within a small graduate program in Health Sciences. All listed authors contributed to the project in meaningful ways, starting from the early stages of conceptualization and project development. Some authors served on the thesis defense committee, others contributed through methodological and analytical support, and several provided feedback on the manuscript draft or the study design. Their inclusion reflects the collaborative and interdisciplinary nature of the research team, which is essential in addressing a complex topic.

Round 2
Reviewer 3 Report
Comments and Suggestions for Authors
Dear authors, I have a few more small suggestions.
What is the reason that the newly added authors were omitted from the original version of the text?
I think that the explanation of the data collection method, as you mentioned in the comment, further contributed to the transparency of the data.
If you are already using R, try making a line graph in R. Try entering points on it that correspond to the values on the x-axis.
Author Response
We would like to sincerely thank you for your careful reading, constructive feedback, and generous dedication throughout this process. Your thoughtful comments have significantly improved the clarity and robustness of our manuscript.
Please accept our apologies for the delay in this final round of responses. We remain at your disposal for any further clarifications or adjustments you may deem necessary.
Comment 1:
What is the reason that the newly added authors were omitted from the original version of the text?
Response 1:
Thank you for raising this point. We would like to clarify that no new researchers were added to the manuscript. The change refers only to a correction in the full names of two co-authors. In the original submission, due to a clerical error, their middle names were omitted. This was brought to our attention after the preprint was posted, and we promptly corrected it to ensure proper indexing and attribution. No changes in authorship have occurred.
Comment 2:
I think that the explanation of the data collection method, as you mentioned in the comment, further contributed to the transparency of the data.
Response 2:
We appreciate the reviewer’s continued interest in the structure of the Brazilian mortality data system. As this is an ecological study based on aggregated public health data, we collected information exclusively from the official government mortality database (SIM/DATASUS), without direct access to individual death certificates or case-level documentation. Unlike field investigations that validate the cause of death on site or by consulting medical/legal records—a process typically restricted to government agencies—our study did not include such procedures due to lack of authorization and operational limitations.
To address the reviewer’s suggestion, we have added a paragraph in the Limitations section explaining how the death certification process works in Brazil, the potential gaps in forensic coverage in rural and Indigenous areas, and the existence of two complementary forms designed to reduce misclassification. These details strengthen the transparency of our data source and acknowledge the possible impact of systemic inconsistencies on data quality.
Comment 3:
If you are already using R, try making a line graph in R. Try entering points on it that correspond to the values on the x-axis.
Response 3:
We apologize for the delay in implementing this suggestion. Although I am the primary author, the co-authors most familiar with the R programming language are not myself, but two of our colleagues. As most members of our team are not exclusively dedicated to academic research—with the exception of Professors Maria Dalva, Sandra, and Raíssa—it took some time to coordinate our schedules and discuss the reviewer’s suggestions in detail. The three-day period proved too short for this purpose.
Nevertheless, we have now prepared three alternative versions of the figure, all created in R, and included them in the supplementary file titled Figure 2 – alternatives. At this point, we have not replaced the original figure in the manuscript because we would like to confirm with the reviewer which version they consider most appropriate. Once we receive this guidance, we will adjust the manuscript accordingly, including aligning the figure caption and corresponding description in the Results section.
Lastly, we would like to explain that our initial choice to present the graph using absolute numbers was motivated by the visual clarity it offers, particularly in identifying temporal trends in non-indigenous and seasonal patterns in the Indigenous population.
